# Who are the patients being offered the faecal immunochemical test in routine English general practice, and for what symptoms? A prospective descriptive study

Natalia Calanzani ,[1] Merel M Pannebakker ,[1] Max J Tagg ,[2]
Hugo Walford ,[3] Peter Holloway,[4] Niek de Wit,[5] Willie Hamilton ,[6]
Fiona M Walter [1,7]

For numbered affiliations see end of article.

**Correspondence to**
Dr Natalia Calanzani; nm719@medschl.cam.ac.uk

## ABSTRACT

**Objectives** The faecal immunochemical test (FIT) was introduced to triage patients with lower-risk symptoms of colorectal cancer (CRC) in English primary care in 2018. While there is growing evidence on its utility to triage patients in this setting, evidence is still limited on how official FIT guidance is being used, for which patients and for what symptoms. We aimed to investigate the use of FIT in primary care practice for lower-risk patients who did not immediately meet criteria for urgent referral.

**Design** A prospective, descriptive study of symptomatic patients offered a FIT in primary care between January and June 2020.

**Setting** East of England general practices.

**Participants** Consenting patients (aged ≥40 years) who were seen by their general practitioners (GPs) with symptoms of possible CRC for whom a FIT was requested. We excluded patients receiving a FIT for asymptomatic screening purposes, or patients deemed by GPs as lacking capacity for informed consent. Data were obtained via patient questionnaire, medical and laboratory records.

**Primary and secondary outcome measures** FIT results (10 µg Hb/g faeces defined a positive result); patient sociodemographic and clinical characteristics; patient-reported and GP-recorded symptoms, symptom severity and symptom agreement between patient and GP (% and kappa statistics).

**Results** Complete data were available for 310 patients, median age 70 (IQR 61–77) years, 53% female and 23% FIT positive. Patients most commonly reported change in bowel habit (69%) and fatigue (57%), while GPs most commonly recorded abdominal pain (25%) and change in bowel habit (24%). Symptom agreement ranged from 44% (fatigue) to 80% (unexplained weight loss). Kappa agreement was universally low across symptoms.

**Conclusion** Almost a quarter of this primary care cohort of symptomatic patients with FIT testing were found to be positive. However, there was low agreement between patient-reported and GP-recorded symptoms. This may impact cancer risk assessment and optimal patient management in primary care.

## STRENGTHS AND LIMITATIONS OF THIS STUDY

⇒ Recent data on how faecal immunochemical test (FIT) is used in primary care, and which symptoms trigger a FIT request.
⇒ To our knowledge, this is the first study in England to report on the agreement between patient-reported and general practitioner-recorded symptoms when a FIT is requested in primary care.
⇒ Limitations include poor access to primary care records, and consequently limited evidence on cancer diagnoses.
⇒ There was over-representation of white British populations; therefore, results may not be generalisable to different groups.

## INTRODUCTION

Colorectal cancer (CRC) is the second most common cause of cancer deaths worldwide and in the UK.[1] Despite screening programmes, most CRCs are diagnosed after symptomatic presentation,[2] with patients usually presenting first in primary care.[3] Those presenting with alarm symptoms equating to a CRC risk of ≥3% can be referred via an urgent suspected cancer pathway, following the National Institute for Health and Care Excellence (NICE) NG12 guidelines.[4] Recognising that patients with lower-risk ('low-risk, but not no-risk')[5] symptoms may also have cancer, more recent NICE guidance (DG30) recommends offering a faecal immunochemical test (FIT) to patients 'without rectal bleeding who have unexplained symptoms but do not meet criteria for a suspected cancer pathway'.[6] Those with results showing blood concentration of ≥10 µg Hb/g faeces are recommended to be referred for further investigations, most often a colonoscopy.

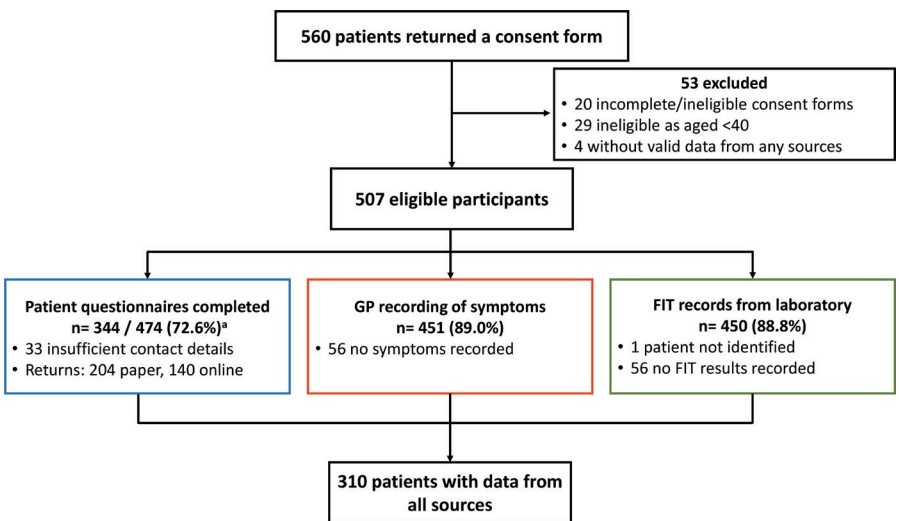

**Figure 1** Study flow diagram. [a]Sixteen patients returned more than one questionnaire. The first response was kept to reduce recall bias. FIT, faecal immunochemical test; GP, general practitioner.

The evidence review that informed DG30 development did not identify any studies using FIT in primary care before referral (ie, as a triage tool),[7] raising the risk of spectrum bias (variation in performance due to testing in different populations).[8] However, since implementation of this guidance, several studies in primary care (including patients with higher-risk and/or lower-risk symptoms)[9–17] have demonstrated that FIT is an effective tool for CRC in primary care, helping to reassure patients and professionals when results are negative, and optimising referrals for definitive investigations when results are positive. Different FIT thresholds may also be used to indicate patients with different levels of risk.[9 10 17] Due to the COVID-19 pandemic, use of FIT has recently been intensified, including to prioritise urgent referrals.[18]

A comprehensive analysis of diagnostic performance of FIT in Southwest England showed that FIT performs exceptionally well to triage patients presenting in primary care with lower-risk CRC symptoms (as specified by DG30 guidance).[9] However, evidence is still limited on how guidance is being used in practice, and for which patients. There is also little evidence about the presenting symptom/s that prompt patients to consult and general practitioners (GPs) to request a FIT. The aim of this study was to investigate use of FIT in English primary care for patients who did not immediately meet criteria for urgent referral. We report on the characteristics of patients for whom a FIT was requested, and agreement between patient-reported and GP-recorded symptoms.

## METHODS
### Design
We conducted a pragmatic, prospective, descriptive study in the East of England, where a quantitative FIT service for diagnostic triage of lower-risk symptomatic patients was launched in primary care from late 2018.

### Setting and population
The study was set in all Eastern Cancer Alliance General Practices in two of six Sustainability and Transformation Plans: Suffolk & North East Essex and Norfolk & Waveney (estimated adult population of 1 160 215).[19]

We included adult patients aged ≥40 years who were seen by their GPs with symptoms of possible CRC between January and June 2020, but who were not urgently referred for investigation (NICE NG12),[4 6] and for whom a FIT was requested. We excluded patients receiving a FIT for asymptomatic screening purposes, or patients deemed by GPs as lacking capacity for informed consent.

Two laboratories responsible for all the FITs added study recruitment letters, consent forms and a prepaid envelope to FIT kits distributed to GP practices. These were given to patients when a FIT was requested for patients who did not immediately meet urgent referral criteria.

### Data sources
#### Study data
Data collection started on 24 January 2020 (kits sent from laboratories to GP practices) and ceased on 31 January 2021 (data obtained from primary care records). Patient consent forms were only accepted until 30 June 2020 due to clinical pathway changes resulting from the COVID-19 pandemic (ie, FIT being increasingly performed in primary care for higher-risk patients who before the pandemic would meet criteria for urgent referral).

Patients were asked about their age, gender and postcode (to measure deprivation - shown in quintiles) on consent forms and invited to complete a questionnaire (online supplemental file 1) by post or online (Qualtrics, Provo, Utah, USA). A reminder was sent after two weeks. The questionnaire was developed from the SYMPTOM questionnaire[20–22] and the Family History Questionnaire.[23 24] It included items on symptoms (change in

**Table 1** Sociodemographic characteristics of sample

| | Full dataset (n=310) |
|---|---|
| Available for all the initial sample | |
| Age | |
| Median age (IQR) | 70 (61–77) |
| Gender | n (%) |
| Female | 160 (53) |
| Male | 142 (47) |
| Deprivation (Index of Multiple Deprivation quintiles) | |
| 1 (least deprived) | 70 (23) |
| 2 | 73 (24) |
| 3 | 88 (28) |
| 4 | 49 (16) |
| 5 (most deprived) | 30 (10) |
| Questionnaire data | |
| Ethnicity | n (%) |
| White British | 289 (94) |
| White non-British* | 13 (4) |
| Other† | 7 (2) |
| Education | |
| Higher education | 110 (37) |
| A-level or equivalent | 33 (11) |
| General Certificate of Secondary Education/O-level or equivalent | 79 (26) |
| Other‡ | 18 (6) |
| None | 60 (20) |
| Family history | |
| Close relatives with CRC before the age of 55 | 17 (66) |
| More than one relative with CRC | 40 (13) |

Percentages may not add up to 100% due to rounding.
Missing data: gender (full dataset n=8, 2.6%; initial dataset n=12, 2.4%), ethnicity (n=1, 0.3%), education (n=11, 3.2%). Percentages in table calculated considering complete cases.
*White Irish (n=7), any other white background (n=6).
†Not further described to protect patient anonymity as n<5 for each.
‡Any other qualification, for example, 'diploma', 'high school', 'police promotion examinations', 'teacher', etc.
CRC, colorectal cancer.

bowel habit, change in bowel habit with mainly diarrhoea, change in bowel habit with mainly constipation, abdominal pain, rectal bleeding, indigestion/heartburn, bloating, fatigue, unexplained weight loss, loss of appetite and mucus in faeces), symptom severity (Likert-type items, from mild (1) to unbearable (6)), sociodemographic characteristics and lifestyle behaviours, medication use, medical history and family history of CRC.

The laboratories also collected FIT and blood test results on consenting patients. The FIT assays used were OC-Sensor (Norfolk) and HM-JACKarc (Suffolk): a 'positive' test was defined as faecal haemoglobin ≥10 µg Hb/g faeces, as per NICE DG30.[6] Results using the OC-Sensor were presented in ng/mL, which were then converted into µg Hb/g by dividing the results by 5. Additionally, laboratories provided GP-recorded symptom data, collected via the FIT request form which had been adapted with the Eastern Cancer Alliance, to match symptoms in the patient questionnaire (except for additional questions about mucus in faeces and symptom severity which were only available for patients).

Finally, primary care records were accessed by practice staff or remotely by the research team (if approved by practices) ≥6 months after FIT was requested to collect clinical outcome data (medical history and cancer diagnoses). These data were only available for one-third of patients as practices had to be contacted about each consenting patient—resulting in unfeasible additional workload for many primary care practices during the COVID-19 pandemic. It is unlikely that these data were missing at random and clinical outcome data are only reported descriptively to avoid misleading results—no measures of diagnostic performance were therefore calculated.

Data on patient age, gender and postcode from consent forms were cross-checked when provided by more than one data source (patient responses were prioritised). A database was created in Microsoft Access V.2016 (secure server, password protected) for data collection and recording of key data, study timelines and information on data completeness.

### Historical data for comparison purposes
Anonymised historical, aggregated data on FIT requests for symptomatic patients, FIT results and GP-recorded symptom/s with FIT request were provided by the North East Essex and Suffolk Pathology Services and the National Health Service Ipswich and East Suffolk Clinical Commissioning Group. These data covered the period from FIT implementation (test launch in primary care) in November 2018 to May 2020, for the same regions on the East of England (Suffolk & North East Essex and Norfolk & Waveney). This was important for comparison purposes, and for assessing relevance of study population/symptomatology.

### Data analysis
Data from consenting patients were anonymised and exported into SPSS V.27 format. Analyses were carried out for patients with data from questionnaires, for whom there were also FIT results and GP-reported symptom data. This ensured a full, complete dataset.

Body mass index was calculated using patient self-reported values (kg/m$^2$). Practice postcodes were used to measure deprivation when patient postcodes were missing. All other missing data were coded as such,

**Table 2** Patient lifestyle, medication use and comorbidities

| | Full dataset (n=310) |
|---|---|
| | n (%) |
| Questionnaire data | |
| Smoking status | |
| Current smoker | 23 (7) |
| Ex-smoker | 125 (40) |
| Non-smoker (never smoked) | 162 (52) |
| Alcohol in last 12 months* | |
| Frequent drinker | 138 (45) |
| Infrequent drinker | 48 (16) |
| Teetotal/rarely drinks | 124 (40) |
| BMI | |
| Underweight (below 18.5) | 12 (4) |
| Normal (18.5–24.9) | 135 (44) |
| Overweight (25–29.9) | 101 (33) |
| Obese (30 and above) | 56 (18) |
| Imaging and screening | |
| Bowel screening test (past 5 years) | 226 (73) |
| Medication use | |
| NSAIDs | 59 (19) |
| Aspirin | 47 (15) |
| Anticoagulants/blood thinners | 51 (17) |
| Antidepressants | 38 (12) |
| None of the above | 145 (47) |
| Comorbidities (non-GI only) | |
| Diabetes | 43 (14) |
| Arthritis | 95 (31) |
| Anxiety or depression | 77 (25) |
| Laboratory data† | |
| Hb‡ (n=282) | |
| Low | 51 (18) |
| Within normal range | 231 (82) |
| Platelet count§ (n=282) | |
| Low | 11 (4) |
| Within normal range | 254 (90) |
| High (>400) | 17 (6) |
| ESR¶ (n=53) | |
| Low | 3 (6) |
| Within normal range | 32 (60) |
| High | 18 (34) |
| CRP** (n=190) | |
| Within normal range | 142 (75) |
| High | 48 (25) |

Continued

**Table 2** Continued

| | Full dataset (n=310) |
|---|---|

Percentages may not add up to 100% due to rounding.
Missing data: BMI (n=6). Percentages in table calculated considering complete cases.
*Frequent drinker: almost every day or about twice a week. Infrequent drinker: about once a week or about once a fortnight. Teetotal/rarely drinks: only a few times a year or never drinks alcohol.
†Four cases with available test results were excluded as they were not carried out ≤6 months of a FIT (n=3) or dates were not available (n=1)—these tests are more likely to have been carried out for different reasons. Median interval between FIT and other tests was 2 days (IQR 0-6.5 days).
‡Hb ranges: low <130 men and <115 women, within normal range 130–180 men and 115–165 women.
§Platelet ranges: low <140, within normal range 140–400, high >400.
¶ESR ranges: low <1 men and <3 women, within normal range 1–10 men and 3–15 women, high >10 men and >15 women.
**CRP within normal range 0–5, high >5.
BMI, body mass index; CRP, C-reactive protein; ESR, erythrocyte sedimentation rate; FIT, faecal immunochemical test; GI, gastrointestinal; Hb, haemoglobin; NSAIDs, non-steroidal anti-inflammatory drugs.

using the same code/label to ensure consistency across different variables and datasets.

Descriptive statistics (frequencies, percentages and measures of central tendency) were used for patient self-reported data, GP-recorded symptoms, symptom severity and test results. Agreement between symptoms reported by patients and recorded by GPs was calculated as the number of agreed upon symptoms (both patient-reported and GP-recorded) divided by the sum of cases with agreements and the cases with disagreements. Kappa statistics were calculated and interpreted as: poor or worse than chance (<0.00), slight (0.00–0.20), fair (0.21–0.4), moderate (0.41–0.60), substantial (0.61–0.80) or almost perfect (0.81–1.00) agreement.[25]

### Patient and public involvement
There was no patient and public involvement (PPI) in this analysis. However, the study also includes a qualitative component to assess views on FIT experience and acceptability (reported separately) with PPI (specifically, reviewing and analysing a subsection of interview transcripts).

### RESULTS
### Population and setting
From January to June 2020, 560 patients from Suffolk & North East Essex and Norfolk & Waveney (average 108 per month) consented to take part in the study.

Historical data (online supplemental file 2) for the first half of 2020 (January–May 2020) show that there were 6287 FIT requests in these regions (average 1257 per month with variation during the COVID-19 pandemic).

## Patient flow

A total of 507 consenting patients (91%) were eligible for inclusion ('initial sample'). Linked data from all sources ('full dataset' with patient data (questionnaire), laboratory records and GP recording of symptoms for each patient) were available for 310 (61%) linked patients (figure 1).

## Patient characteristics

In the full dataset (n=310), almost a quarter of FIT results were positive (23%, n=72). Historical data (same regions in Eastern England) show a similar proportion of positive FITs from November 2018 (test launch) to May 2020 (15 620 valid FITs were performed overall, 21% (n=3237) positive) (online supplemental file 2).

Patient median age was 70 (IQR 61–77) years and over half were female (53%) (table 1). About a quarter of patients (26%) lived in areas of higher deprivation. Compared with the initial sample (n=507), the full dataset was representative in terms of gender, but the most deprived and younger patients were under-represented (online supplemental table 1). Most patients identified as white British (94%), over one-third had a higher education degree (37%) and two-thirds (66%) reported having had a close relative with CRC prior to the age of 55 years.

Table 2 shows that half the sample were non-smokers, 45% were frequent drinkers, over 50% were either overweight or obese, and almost three-quarters had a bowel screening test in the past 5 years. The most common non-gastrointestinal comorbidities were arthritis and anxiety or depression, and most blood test results were normal.

## Symptoms reported by patients and recorded by their GPs
### Commonly reported symptoms

The most common patient-reported symptoms were change in bowel habit (69%) and fatigue (57%), while the least common was rectal bleeding (23%). In comparison, the most common GP-recorded symptoms were abdominal pain (25%) and change in bowel habit (24%), while the least common was loss of appetite (3%) (table 3). Two hundred different patient-reported symptom patterns were identified; most of them (n=140) were reported only once. The maximum number of times a symptom pattern was reported was 9 times (fatigue only), or 11 times if having no symptoms is considered a pattern (online supplemental file 3).

Historical data for the same regions in Eastern England also show that abdominal pain (20%) and change in bowel habit (35%) were the most common GP-recorded symptoms, while loss of appetite was also the least common (3%) (online supplemental file 2).

## Patient and GP agreement

Percentage agreement between patient-reported and GP-recorded symptoms ranged from 44% for fatigue to 80% for unexplained weight loss. Kappa agreement was universally low across symptoms (lowest for bloating, fatigue and loss of appetite). GPs recorded symptoms less often, and recorded fewer symptoms per patient (median 1 symptom, IQR 1–1), compared with patients (median 5, IQR 3–6—excluding mucus in faeces) (table 3).

Similarly, historical data for the East of England show that the proportion of GP-recorded symptoms was lower when compared with patient-reported symptoms, for all symptoms (online supplemental file 2).

## Patient-reported symptom severity

Patients reported that most symptoms (except for change in bowel habit and change in bowel habit: diarrhoea) were usually either mild or moderate. 'Severe' and 'very severe' symptoms were reported for all types of change in bowel habit more frequently than for other symptoms (figure 2).

The proportion of GPs recording symptoms often increased based on patient-reported symptom severity. This was particularly evident for rectal bleeding, abdominal pain and unexplained weight loss, although there was variation due to small numbers (particularly very few 'very severe' or 'unbearable' cases) (figure 2).

## Clinical outcomes

Clinical outcomes were only available for 105 patients in the full dataset (34%); 26 had a positive FIT result (25%). Three patients received a CRC diagnosis (3% of 105 patients and 12% of FIT positives), comprising two rectal cancers and one carcinoma of the caecum. All CRC cases had positive FIT results with values >400 µg Hb/g faeces. One patient was diagnosed with endometrial cancer (negative FIT result). Twenty-two patients received an alternative gastrointestinal diagnosis, most commonly diverticular disease, diverticulosis or diverticulitis (n=12).

## DISCUSSION

This study set in routine English primary care showed that FIT usage for symptomatic patients was greater in women and in older patients—perhaps unsurprisingly as CRC incidence increases with age, but the gender balance is different than expected as CRC is more common in men.[1] Patients reported on more symptoms than GPs recorded when ordering FITs. While GP reporting of symptoms increased based on symptom severity, agreement between patient and GP reporting was low suggesting that risk assessment algorithms such as the Risk Assessment Tools (RATs) [26] could underperform. Pre-pandemic, we aimed to understand real-life use of FIT for patients presenting in primary care with symptoms that could indicate CRC, particularly those with lower-risk symptoms. As the COVID-19 pandemic progressed, it became clear that FIT was being used more extensively for patients with

**Table 3** Patient-reported and GP-recorded symptoms and agreement

| | Full dataset (n=310) | | | | | |
|---|---|---|---|---|---|---|
| | **Patient reported*** | **GP recorded†** | **Reported by patient but not recorded by GP** | **Recorded by GP but not reported by patient** | **Agreement** | **Kappa (95% CI)‡** |
| | **n (%)** | **n (%)** | **n (%)** | **n (%)** | **%** | |
| Change in bowel habit (CIBH) | 213 (69) | 75 (24) | 146 (47) | 8 (3)§ | 50 | 0.167 (0.100 to 0.234) |
| CIBH: diarrhoea | 139 (45) | 72 (23) | 77 (25) | 10 (3)¶ | 72 | 0.406 (0.312 to 0.500) |
| CIBH: constipation | 93 (30) | 32 (10) | 69 (22) | 8 (3)** | 75 | 0.272 (0.164 to 0.380) |
| Abdominal pain | 170 (55) | 77 (25) | 114 (37) | 21 (7) | 57 | 0.169 (0.080 to 0.257) |
| Rectal bleeding | 72 (23) | 21 (7) | 60 (19) | 9 (3) | 78 | 0.171 (0.057 to 0.285) |
| Indigestion/heartburn | 128 (41) | 11 (4) | 119 (38) | 2 (1) | 61 | 0.069 (0.014 to 0.124) |
| Bloating | 170 (55) | 16 (5) | 158 (50) | 4 (1) | 48 | 0.038 (0.005 to 0.081) |
| Fatigue | 178 (57) | 35 (11) | 158 (50) | 15 (5) | 44 | −0.001 (−0.064 to 0.062) |
| Unexplained weight loss | 76 (25) | 23 (7) | 57 (18) | 4 (1) | 80 | 0.305 (0.187 to 0.423) |
| Loss of appetite | 89 (29) | 8 (3) | 85 (27) | 4 (1) | 71 | 0.037 (−0.0305 to 0.100) |
| Mucus in faeces | 85 (28) | N/A | N/A | N/A | N/A | N/A |

Percentages may not add up to 100% due to rounding.
*Median interval between FIT request being processed by the laboratory and the patient returning the questionnaire where symptoms were reported was 42 days (IQR 30–56).
†GPs recorded symptoms when requesting a FIT.
‡Kappa interpretation: poor or worse than chance (<0.00): slight (0.00–0.20): fair (0.21–0.4), moderate (0.41–0.60), substantial (0.61–0.80) or almost perfect (0.81–1.00) agreement.
§In all these cases, patient had not reported change in bowel habit, diarrhoea nor constipation.
¶In five of these cases, patient had reported change in bowel habit.
**In six of these cases, patient had reported change in bowel habit.
FIT, faecal immunochemical test; GP, general practitioner; N/A, not applicable.

higher-risk symptoms, therefore we stopped the study prematurely.

This study provides much needed data on agreement between patient-reported and GP-recorded symptoms prompting a FIT request. To our knowledge, this is the first time that these findings are reported for English populations. While this study was not powered to look at cancer outcomes, poor access to primary care records compounded this aspect, and was a key limitation. The analytical approach, creating a full dataset that contained key variables for all patients (symptom data and FIT results), reduced our sample size, with an over-representation of the least deprived and older patients. The study also over-represented the white British population (estimated to be 85.2% in the East of England according to the latest Census).[27] Nonetheless, comparisons with historical data from the East of England showed similar patterns regarding the proportion of positive FITs, and on GP-recorded symptoms. These similarities further validate the relevance of this study. Finally, patients did not provide information on symptoms at the same time as GPs (increasing the risk of recall bias) and different symptom collection methods were used.

Under-recording of symptoms by healthcare professionals is common and has also been previously linked to symptom severity and their perceived importance.[28–33] We identified only one other study describing both patient-reported and GP-reported symptoms when a FIT is requested in primary care.[33] Set in Swedish primary care, Högberg *et al* also described low symptom agreement, with GPs reporting symptoms less often.[33] In our study, GPs were compelled to tick one or more symptoms from a list to request a FIT; GPs may have chosen to record only the symptoms that triggered clinical action, and/or prioritised what to record. Their clinical interpretation of a symptom (such as its importance or duration) may also differ from the patient perspective. This may explain why GPs reported symptoms which sometimes were not described by patients (often alarm symptoms such as abdominal pain, rectal bleeding and change in bowel habit), and why agreement was highest for unexplained weight loss and rectal bleeding, both alarm symptoms.

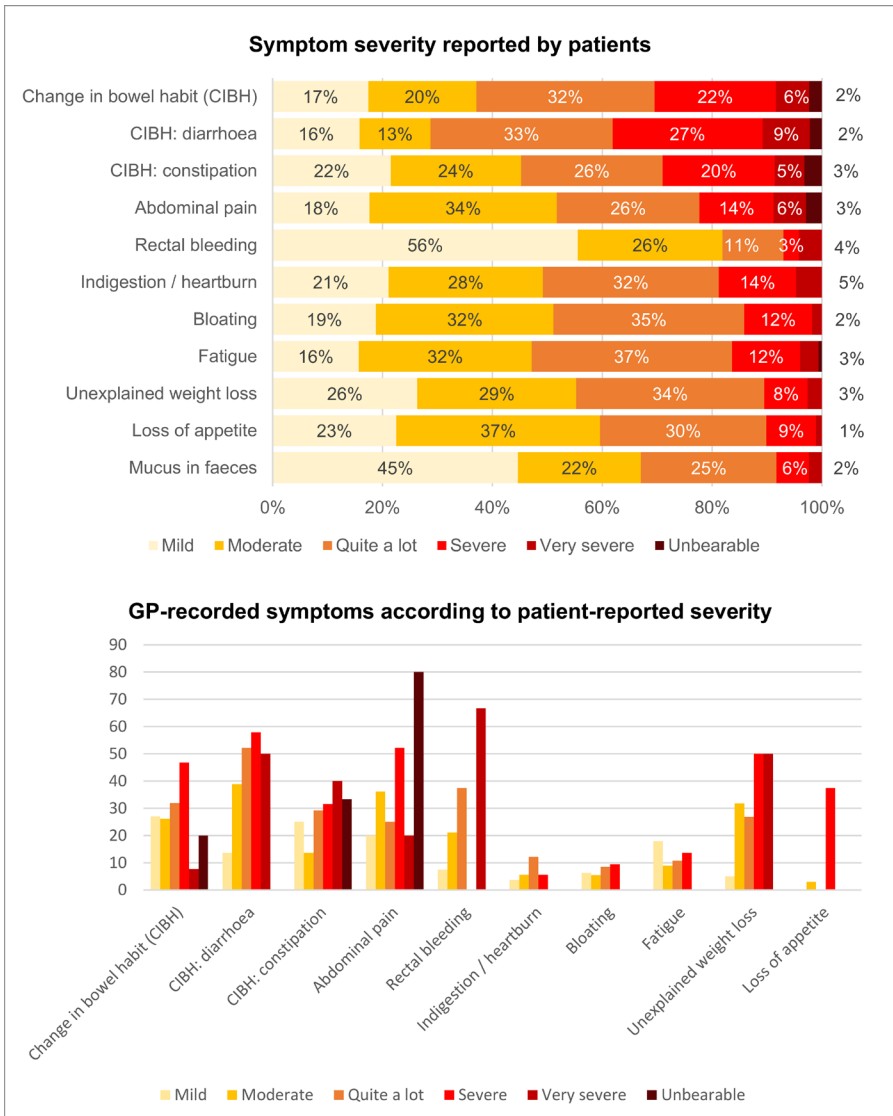

**Figure 2** Symptom severity. GP, general practitioner.

Some patients may have under-reported their symptoms to their GP, or only mentioned the most severe symptom. Nonetheless, low symptom agreement between GPs and patients suggests possible recording bias and suboptimal record-keeping. These can have implications for patient management, such as the risk of underestimating the importance of certain symptoms for patients' cancer risk assessment and quality of life, although this possibility needs to be further investigated. It is vital that GPs are aware of this discrepancy in symptom reporting/ recording; our findings could help to inform education materials for GPs and other healthcare professionals. Improving accuracy of symptom recording could help GPs to better understand the symptom continuum (ie, symptoms that persist or worsen over time) and improve efficacy of risk assessment tools embedded in electronic medical records. This not only has crucial implications for detecting cancers in primary care,[34] but can also more accurately guide detection of conditions other than cancer.

The Swedish study found the most common patient-reported symptoms were abdominal pain and diarrhoea, in contrast to change in bowel habit and fatigue in our study.[33] Both studies reported the same two most common GP-reported symptoms, abdominal pain and change in bowel habit. These symptoms, alongside age criteria, could potentially trigger an urgent referral in the UK.[4] It may be that GPs assessed these patients as being lower risk, based on other clinical characteristics.[11 15] Our results suggest that FIT is being used to aid assessment of 'grey area' patients, in line with its initially intended use. However, the high proportion of reported alarm symptoms, alongside older populations, also indicates that FIT was being used beyond its intended use, including for patients previously considered to be higher risk. This was partly due to new guidance brought in response to the COVID-19 pandemic but cannot wholly be explained by this public health emergency.

Importantly, this wider adoption of FIT for patients with both lower-risk and higher-risk symptoms does not

mean that the more recent use is not appropriate. Recent studies do show that FIT has the potential to be used more widely.[9 10 17] Furthermore, recent evidence of the expanded use of FIT in England during the COVID-19 pandemic shows a range of positive outcomes, including reduction in mortality,[18] significant reduction in the use of endoscopy without compromising CRC detection rates[35] and successful prioritisation of patients with a higher risk of cancer when endoscopy services were severely restricted.[36] GPs also had positive views on the use of FIT during the COVID-19 pandemic.[37] Finally, there is evidence that FIT is seen as an acceptable test by symptomatic patients.[38] Perhaps FIT use has become more embedded and widespread in primary care, and clinicians have become more confident with its use(s). With access to hospital services including endoscopy under pressure during the pandemic, FIT has become a standard filter triage test in vague symptom pathways and is now an integral part of most two-week wait colorectal pathways.[39 40] Furthermore, recent guidance from the Association of Coloproctology of Great Britain & Ireland and the British Society of Gastroenterology recommends the use of FIT as a diagnostic triage tool in primary care, including for patients with red flag symptoms such as rectal bleeding.[41 42] NICE also plans to publish additional guidance on FIT by November 2023.[43] There is also ongoing work being conducted to support the introduction of using FIT results to risk stratify patients into a three-tier referral system for full colonoscopy, capsule endoscopy and no further investigation.[44] Future research will be required to assess this expanding role of FIT outside the current NICE referral guidelines. With an increasing reliance on FIT as a triage tool for symptomatic patients in primary care, particular attention should be given to the application of repeat FIT (with optimum intervals and thresholds)[45] and the impact of using different cut-off points[33]: safety-netting will become even more vital to avoid missing cancer cases with a negative FIT result.[45–47]

## Author affiliations
[1]The Primary Care Unit, Department of Public Health and Primary Care, University of Cambridge, Cambridge, UK
[2]School of Clinical Medicine, Addenbrooke's Hospital, University of Cambridge, Cambridge, UK
[3]School of Clinical Medicine, University College London, London, UK
[4]East of England Cancer Alliances, Cambridge, UK
[5]Julius Center for Health Sciences and Primary Care, University Medical Center, Utrecht University, Utrecht, The Netherlands
[6]University of Exeter Medical School, University of Exeter, Exeter, UK
[7]Wolfson Institute of Population Health, Queen Mary University of London, London, UK

**Acknowledgements** We thank all the patients who agreed to take part in the study, and who completed the questionnaire. We thank the general practice staff who provided us with patient outcome data. We thank Allison Chipchase, Eve Calderbank and Claire Corbett for their help obtaining historical data. We thank the North East Essex and Suffolk Pathology Services at the East Suffolk and North Essex NHS Foundation Trust, and the Clinical Biochemistry section of Laboratory Medicine at Norfolk and Norwich University Hospitals NHS Foundation Trust for all their help providing laboratory data. We are grateful to Dr Marije van Melle for her work in the early stages of study design, and to Dr Sarah ER Bailey for her help during protocol development. We are grateful to Dr Lina Massou for expert statistical input. We thank Aina Chang and Sara Shaida for their help with data collection and gathering background information for the study, and James Brimicombe for data management advice, support and expertise. We are indebted to Andy Cowan for his invaluable support collecting and managing study data, and for his comments on the study results. We also thank Victoria Hardy, Dr Owain T Jones and Dr Yin Zhou for their comments on the study results.

**Contributors** MMP and FMW planned the original study. WH and FMW secured funding. Analysis was planned by MMP, NC, MJT, HW, NdW, WH, PH and FMW, and carried out by NC, MJT, HW and FMW. NC drafted the manuscript with senior input from WH, NdW and FMW, and revised it after critical reviews from all authors. All authors read and approved the final manuscript. NC is acting as the guarantor.

**Funding** This research arises from the CanTest Collaborative, which is funded by Cancer Research (UK C8640/A23385), of which FMW and WH are Directors, NdW is an Associate Director, NC and MMP are postdoctoral researchers.

**Disclaimer** The funders of the study had no role in study design, data collection, data analysis, data interpretation or writing of the report.

**Competing interests** None declared.

**Patient and public involvement** Patients and/or the public were not involved in the design, or conduct, or reporting, or dissemination plans of this research.

**Patient consent for publication** Not required.

**Ethics approval** This study involves human participants and ethical approval was granted by the East of England Cambridge Central Research Ethics Committee (reference 19/EE/0036). Participants gave informed consent to participate in the study before taking part.

**Provenance and peer review** Not commissioned; externally peer reviewed.

**Data availability statement** Data are available upon reasonable request.

**ORCID iDs**
Natalia Calanzani http://orcid.org/0000-0002-5068-2543
Merel M Pannebakker http://orcid.org/0000-0003-2918-0570
Max J Tagg http://orcid.org/0000-0003-0341-3284
Hugo Walford http://orcid.org/0000-0001-9655-9674
Willie Hamilton http://orcid.org/0000-0003-1611-1373
Fiona M Walter http://orcid.org/0000-0002-7191-6476

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
