## [Reviewer comments · BMJ Open]

ARTICLE DETAILS

TITLE (PROVISIONAL)	Who are the patients being offered the faecal immunochemical test in routine English general practice, and for what symptoms? A prospective descriptive study
AUTHORS	Calanzani, Natalia; Pannebakker, Merel M; Tagg, Max J; Walford, Hugo; Holloway, Peter; de Wit, Niek; Hamilton, Willie; Walter, Fiona M

VERSION 1 – REVIEW

REVIEWER	Joachim Worthington The University of Sydney, Daffodil Centre
REVIEW RETURNED	08-Jul-2022

GENERAL COMMENTS	This is a strong and clear paper on the use of FIT to triage patients presenting to primary care with low-risk possible symptoms colorectal cancer symptoms. The use of FIT to manage lower-risk patients has been increasingly important, both due to the increasing prevalence of colorectal cancer and subsequent pressure on colonoscopy demand, and due to the health system limitations presented by the COVID-19 pandemic. I believe the manuscript would benefit significantly from an expanded discussion of the context of COVID, which had a big impact on this study. For obvious reasons the study was not designed to address this, but the same reasoning that meant the study was completed prematurely would also be a clear demonstration of the importance of this study - the increasing reliance on FIT as a primary care triage tool. You note that FIT is being used "beyond its [currently] intended use" in the discussion - I think it is worth noting that although this is beyond current recommendations, it is possible that this broader use of FIT may be found to be appropriate, especially during COVID time. Of course, there is not evidence to support this either way, but it is already happening in practice, and this study provides a very nice illustration of what this may look like and what areas may be worth exploring further. I also think stronger and clearer conclusions could be made from the low patient/clinician symptom reporting agreement, both in the abstract and the main text. This is a big focus of your analysis, and seems quite novel to me. However, as written, your conclusion is quite broad - I would've liked to see specific possibilities and examples of how practice could be improved by better capturing relevant symptoms on the clinical side. For example, perhaps this could be used to identify lower risk patients at subsequent follow-ups based on previous FIT negative results, even if they have recurrent symptoms? I am sure you can think of examples, but I definitely feel
---

it would improve the manuscript if the reader's imagination can be sparked on the possible interpretation of this very interesting aspect of your study.

It should also be made clearer throughout, including in the abstract and possibly title, that the focus of the paper is specifically on FIT as triage for low-risk symptomatic patients - not FIT as screening for asymptomatic patients (as in most "organised screening" programs), nor as a replacement for colonoscopy in high-risk patients. You note in the paper that this is a fairly specific cohort with "in-between" risk, and I think this needs to be emphasised.

I think this is a very strong and interesting study - although it is not an explicit outcome of your study, the reader has the impression that the use of FIT as triage in this case is very successful. A 23% FIT positive rate is much higher than in an asymptomatic population (as one would expect), and equally useful in identifying the 77% for whom a colonoscopy may not be useful. This positivity rate seems like a very good "Goldilocks" figure for this patient group - not asymptomatic, but not high enough risk to immediately trigger a colonoscopy. Although this is hard to quantify, I think this study provides a great background to the clinical use of FIT for this purpose. I recommend this study for publication, after minor revisions to make the context and impact of this study clearer based on the comments above.

Some specific notes:

Page 3

Line 6: I think this could be clearer that FIT was introduced _for this purpose_ in 2018 - perhaps changing order to "The faecal... ..in English primary care in 2018".

Line 27: Could be clearer around "screening purposes" - either "routine" or "asymptomatic" screening

Page 5

Line 18: Clarify "results" is "blood concentration", and specify what "further investigations" may entail

Line 30: "reassure" or diagnose, if found to be positive

Line 34: I did not understand this reference to different FIT thresholds for different "levels of risk" - how would this risk be identified? Surely the FIT concentration itself would be the clearest measure of risk level? Can you please clarify this?

Line 60: Please clarify "FIT for diagnostic triage"

Page 6

Line 13: I think this point, that patients who are urgently referred to colonoscopy etc were excluded, is worth emphasising throughout

Line 57 (and a few other places): "consented patients" reads strangely to me - I am unsure if this is an idiosyncrasy I'm unfamiliar with, or if it should be "consenting"

Page 7:

Line 25: I think this is reasonable, due to the very specific cohort in question. It is good that you address this in the discussion as a limitation, though, and I am hopeful that more research could address this in future to better assess the "value" of this approach.

Page 8

Line 16: Suggest reporting $Kappa < 0$ as "worse-than-chance" agreement or similar, instead of just "poor" - I think this is an important distinction to make it a little clearer how to interpret these

	statistics. Page 9: Line 9: I am not sure where this 24% is coming from - is this the top 2 quintiles? Unless I'm missing something, this doesn't appear to be 24% - can you check? Line 11: can you specify what "most characteristics" includes - seems to exclude ethnicity? Page 11: Table 3: are the bolded Kappa values the "statistically significant" agreement, or the "not statistically significant" agreement? Difficult to interpret this. Page 12: Line 32: Was there any data on precancerous polyps + polypectomies in this group?
--	--

REVIEWER	Smita Bakhai University at Buffalo
REVIEW RETURNED	25-Jul-2022

GENERAL COMMENTS	Excellent topic. Great work! Need review by biostat. Minor revision may be needed based on this special review.
---

VERSION 1 – AUTHOR RESPONSE

Reviewer 1	Authors' responses
This is a strong and clear paper on the use of FIT to triage patients presenting to primary care with low-risk possible symptoms colorectal cancer symptoms. The use of FIT to manage lower-risk patients has been increasingly important, both due to the increasing prevalence of colorectal cancer and subsequent pressure on colonoscopy demand, and due to the health system limitations presented by the COVID-19 pandemic.	Thank you for your comments.
I believe the manuscript would benefit significantly from an expanded discussion of the context of COVID, which had a big impact on this study. For obvious reasons the study was not designed to address this, but the same reasoning that meant the study was completed prematurely would also be a clear demonstration of the importance of this study - the increasing reliance on FIT as a primary care triage tool. You note that FIT is being used "beyond its [currently] intended use" in the discussion - I think it is worth noting that although this is beyond current recommendations, it is possible that this broader use of FIT may be found to be appropriate, especially during COVID time. Of course, there is not evidence to support this either way, but it is already happening in practice, and this study provides a very nice illustration of what this may look like and what areas may be worth exploring further.	Thank you for your suggestions. We have added further information on the context of COVID, and of known outcomes when using FIT in this context (last paragraph in the discussion). We have also added a sentence on possible implications of this increased use of, and reliance on, FIT to triage patients in primary care. The added sentences were: "Furthermore, recent evidence of the expanded use of FIT in England during the COVID-19 pandemic show a range of positive outcomes, including reduction in mortality¹⁸, significant reduction in the use of endoscopy without compromising CRC detection rates³⁶, and successful prioritisation of patients with a higher risk of cancer when endoscopy services were severely restricted³⁷. GPs also had positive views on the use of FIT during the COVID-19 pandemic³⁸. Finally, there is evidence that FIT is seen as an acceptable test by symptomatic patients³⁹".

Reviewer 1	Authors' responses
	"With an increasing reliance on FIT as a triage tool for symptomatic patients in primary care, particular attention should be given to the application of repeat FIT (with optimum intervals and thresholds)⁴⁶ and the impact of using different cut-off points³⁴: safety-netting will become even more vital to avoid missing cancer cases with a negative FIT result⁴⁶⁻⁴⁸"
I also think stronger and clearer conclusions could be made from the low patient/clinician symptom reporting agreement, both in the abstract and the main text. This is a big focus of your analysis, and seems quite novel to me. However, as written, your conclusion is quite broad - I would've liked to see specific possibilities and examples of how practice could be improved by better capturing relevant symptoms on the clinical side. For example, perhaps this could be used to identify lower risk patients at subsequent follow-ups based on previous FIT negative results, even if they have recurrent symptoms? I am sure you can think of examples, but I definitely feel it would improve the manuscript if the reader's imagination can be sparked on the possible interpretation of this very interesting aspect of your study	Thank you, we have added further discussions on this (3rd paragraph in the Discussion): "It is vital that GPs are aware of this discrepancy in symptom reporting/recording; our findings could help to inform education materials for GPs and other health care professionals. Improving accuracy of symptom recording could help GPs to better understand the symptom continuum (i.e. symptoms that persist or worsen over time) and improve efficacy of risk assessment tools embedded in electronic medical records. This not only has crucial implications for detecting cancers in primary care³⁵, but can also more accurately guide detection of conditions other than cancer."
It should also be made clearer throughout, including in the abstract and possibly title, that the focus of the paper is specifically on FIT as triage for low-risk symptomatic patients - not FIT as screening for asymptomatic patients (as in most "organised screening" programs), nor as a replacement for colonoscopy in high-risk patients. You note in the paper that this is a fairly specific cohort with "in-between" risk, and I think this needs to be emphasised.	Thank you for your suggestions. While our initial aim was to investigate only low-risk symptomatic patients, it became clear over time that FIT was being used beyond its intended guidance, particularly due to the constraints during the COVID-19 pandemic. We have now made it clearer that the study was about symptomatic (as opposed to asymptomatic) patients, and that our initial aim was to include patients who did not immediately meet criteria for urgent referral (.e. those lower-risk patients). However, we have kept other sections broader (i.e. referring to symptomatic patients, irrespective of risk) to better represent the patients who ended up being included in the study. We trust that this is appropriate. Minor changes were made to the title, abstract (design section), aims (last paragraph in the background section), methods (Design and Historical data for comparison purposes sections), and discussion (first and last paragraphs)
I think this is a very strong and interesting study - although it is not an explicit outcome of your study, the reader has the impression that the use of FIT as triage in this case is very successful. A 23% FIT positive rate is much higher than in an asymptomatic population (as one would expect), and equally useful is identifying the 77% for whom a colonoscopy may not be useful. This	Thank you; we agree that the study is another good example of the use of FIT for symptomatic patients, although we were limited by the poor availability of outcome data after a positive test result. The identified proportion of positive FIT results was also very similar to the proportion identified

Reviewer 1	Authors' responses
positivity rate seems like a very good "Goldilocks" figure for this patient group - not asymptomatic, but not high enough risk to immediately trigger a colonoscopy. Although this is hard to quantify, I think this study provides a great background to the clinical use of FIT for this purpose. I recommend this study for publication, after minor revisions to make the context and impact of this study clearer based on the comments above.	for the South East of England (21%), which further validates the relevance of this study. We have added a sentence to the second paragraph in the discussion emphasising this, i.e: "These similarities further validate the relevance of this study".
Some specific notes: Page 3, Line 6: I think this could be clearer that FIT was introduced _for this purpose_ in 2018 - perhaps changing order to "The faecal... ..in English primary care in 2018".	This has now been amended as suggested.
Page 3, Line 27: Could be clearer around "screening purposes" - either "routine" or "asymptomatic" screening	We have now amended as suggested; i.e. asymptomatic screening purposes
Page 5 Line 18: Clarify "results" is "blood concentration", and specify what "further investigations" may entail	This has now been amended as suggested.
Page 5 Line 30: "reassure" or diagnose, if found to be positive	This has been amended as suggested.
Page 5 Line 34: I did not understand this reference to different FIT thresholds for different "levels of risk" - how would this risk be identified? Surely the FIT concentration itself would be the clearest measure of risk level? Can you please clarify this?	Thank you for your comment; we agree the sentence was confusing. We have now revised it, as the thresholds are indeed the ones used to show different levels of risk. The sentence now reads: "Different FIT thresholds may also be used to indicate patients with different levels of risk"
Page 5, Line 60: Please clarify "FIT for diagnostic triage"	We have amended this as suggested.
Page 6, Line 13: I think this point, that patients who are urgently referred to colonoscopy etc were excluded, is worth emphasising throughout Line 57 (and a few other places): "consented patients" reads strangely to me - I am unsure if this is an idiosyncrasy I'm unfamiliar with, or if it should be "consenting"	Thank you; we have added this information in other sections of the manuscript - as described when addressing the fourth comment. We have changed "consented" to "consenting" as suggested.
Page 7, Line 25: I think this is reasonable, due to the very specific cohort in question. It is good that you address this in the discussion as a limitation, though, and I am hopeful that more research could address this in future to better assess the "value" of this approach.	Thank you; we agree that we need further studies to better understand the usefulness of the use of FIT outwith the already established referral guidelines, and emphasised the need for this in the last paragraph of the manuscript.
Page 8 Line 16: Suggest reporting Kappa<0 as "worse-than-chance" agreement or similar, instead of just "poor" - I think this is an important distinction to make it a little clearer how to interpret these statistics.	We have now added this throughout the manuscript, whenever we mention that there was poor agreement. We have kept the term "poor" as well to be consistent with the adopted Landis and Koch guidelines for Kappa interpretation.
Page 9: Line 9: I am not sure where this 24% is coming from - is this the top 2 quintiles? Unless I'm missing something, this doesn't appear to be 24% - can you check?	Thank you for noticing this; it is a typo and it should have been 26% (16% in the 4 th and 10% in the 5 th quintile). We have now corrected the sentence.
Page 9: Line 11: can you specify what "most	We can see that our text is confusing. Our analysis in supplementary table 1 only included

Reviewer 1	Authors' responses
characteristics" includes - seems to exclude ethnicity?	age, gender and deprivation as these characteristics were available for all patients irrespective of data sources (i.e. questionnaire, GP recording of symptoms and FIT records from laboratory – see Figure 1 for further information). Therefore, our analysis compared these characteristics for those with available (n=310) and not available (n=197) linked data. From this analysis, we know that the most deprived and older patients were underrepresented, but distribution was similar for gender. Data on ethnicity, family history and education were available only for patients who returned the questionnaire (n=344) so this is a different denominator and we did not include these characteristics in the analysis shown in supplementary table 1. The full dataset, with data from all sources, included 310 (90.1%) out of 344 patients who returned the questionnaire. If we compare these patients, i.e. patients with a questionnaire included in the full dataset (n=310) versus patients with a questionnaire not included in the full dataset (n=34), there are no statistically significant differences for ethnicity (Fishers' Exact Test , p=0.483), education (U=5426.5, p=0.353), and having more than one relative with CRC ($\chi(1)=0.595$, p=0.293). On the other hand, there was an underrepresentation of patients with close relatives with CRC before the age of 55 ($\chi(1)=7.265$, p=0.017). Because of the differences in denominator, we think that adding information about this may confuse the reader. Therefore, we have changed the text instead, focusing solely on the comparisons shown in Supplementary Table 1. We have also added a sentence describing the additional patient characteristics: “Most patients identified as White British (94%), over a third had a Higher Education degree (37%) and two-thirds (66%) reported having had a close relative with CRC prior to the age of 55.”
Page 11: Table 3: are the bolded Kappa values the "statistically significant" agreement, or the "not statistically significant" agreement? Difficult to interpret this.	We agree this is difficult to interpret. Significant p-values in kappa indicate that agreement is significantly different from what would be achieved by chance. However, the poor agreement on its own also indicates agreement worse than chance. We have removed the bold formatting as we think it is not adding anything to the analysis.
Page 12: Line 32: Was there any data on precancerous polyps + polypectomies in this group?	Unfortunately the information we have is limited; we know that eight biopsies were carried out among the 105 patients for whom we have clinical outcomes. For the three cancer cases, there was one adenocarcinoma, one high-grade

Reviewer 1	Authors' responses dysplasia in the rectal region, and one lesion was not specified. There were also three benign polyps (one hyperplastic), and in two cases no abnormalities were reported. We opted for not adding this information, alongside information on colonoscopies (n=36), CT scans (n=24) and flexible sigmoidoscopies (n=4), as information was only available for a minority of patients and we wished to focus on patient symptoms.
Reviewer 2	
Excellent topic. Great work! Need review by biostat. Minor revision may be needed based on this special review.	Thank you for your comments. We can confirm that we sought help from an experienced statistician (Dr Lina Massou) and have acknowledged her support in the manuscript.

VERSION 2 – REVIEW

REVIEWER	Joachim Worthington The University of Sydney, Daffodil Centre
REVIEW RETURNED	12-Aug-2022
GENERAL COMMENTS	I am confident that the previous comments have been addressed thoroughly and clearly, and now recommend accepting this manuscript without the need for further revisions.